# Research on Photoinduction-Based Technology for Trapping Asian Longhorned Beetle (*Anoplophora glabripennis* (Motschulsky, 1853) (Coleoptera: Cerambycidae)

**DOI:** 10.3390/insects14050465

**Published:** 2023-05-15

**Authors:** Xianglan Jiang, Xiaoxia Hai, Yongguo Bi, Feng Zhao, Zhigang Wang, Fei Lyu

**Affiliations:** Key Laboratories for Germplasm Resources of Forest Trees and Forest Protection of Hebei Province, College of Forestry, Hebei Agricultural University, Baoding 071000, China; jiangxl710@163.com (X.J.);

**Keywords:** Asian longhorned beetle, pest control, phototactic behavior, violet light, light wavelength, diel rhythm

## Abstract

**Simple Summary:**

The Asian longhorned beetle (*Anoplophora glabripennis*) is an invasive species that attacks at least 209 species (or cultivars) of healthy trees. Several sex pheromones and host kairomones have been used as bites to monitor *A. glabripennis* in the field, but two beetles were captured by the dozens of traps at Cornuda, Italy, while no beetles were captured at Paddock wood, UK. Semiochemical-based traps have still not reached operational efficacy in the field. Therefore, an effective tool is needed for monitoring this beetle. Light traps were widely used in the monitoring and management of pest populations. However, the phototactic behavior of adults remains enigmatic. To provide a theoretical foundation to select the suitable light emitting diode (LED)-based light sources used for monitoring *A. glabripennis*, we first investigated the influence of exposure time and diel rhythm on the phototactic behavior of females and males to provide test conditions, and then tested effect of 14 different wavelength lights and intensity of the most preferred wavelength. Our findings show that 420 nm and 435 nm are the most suitable wavelengths for attracting adult *A. glabripennis* at night. This study can provide a theoretical basis for developing monitoring technologies for *A. glabripennis* based on LED light traps.

**Abstract:**

Light traps play a crucial role in monitoring pest populations. However, the phototactic behavior of adult Asian longhorned beetle (ALB) remains enigmatic. To provide a theoretical foundation to select the suitable light emitting diode (LED)-based light sources used for monitoring ALB, we compared the effect of exposure time on the phototactic response rates of adults at wavelengths of 365 nm, 420 nm, 435 nm, and 515 nm, and found that the phototactic rate increased gradually when the exposure time was prolonged, but there was no significant difference between different exposure times. We evaluated the effect of diel rhythm and found the highest phototactic rate at night (0:00–2:00) under 420 nm and 435 nm illumination (74–82%). Finally, we determined the phototactic behavioral response of adults to 14 different wavelengths and found both females and males showed a preference for violet wavelengths (420 nm and 435 nm). Furthermore, the effect of the light intensity experiments showed that there were no significant differences in the trapping rate between different light intensities at 120 min exposure time. Our findings demonstrate that ALB is a positively phototactic insect, showing that 420 nm and 435 nm are the most suitable wavelengths for attracting adults.

## 1. Introduction

Due to the globalization of international trade and climate change, insect pest outbreaks have become widespread and frequent and can seriously damage agricultural and forestry ecosystems [1]. Chemical pesticides are one of the most commonly used protection methods, but excessive pesticide use over a long period of time has led to many problems, such as reductions in biodiversity, bioamplification of toxic substances within the food web, pest resistance to pesticides, and pollution of the environment [2,3]. Accurate monitoring of insect pest population density is crucial for the successful application of integrated pest management (IPM) strategies because effective and reliable monitoring can provide detailed and timely information on pest population density, thereby reducing the use of chemical pesticides [4], and promoting the timely application of environmentally friendly strategies [5].

The Asian longhorned beetle, *Anoplophora glabripennis*, is ranked as one of the top 100 worst invasive species worldwide and has become established in several places outside of its native range [6,7]. In the United States, if the potential economic losses were adjusted to 2021 values, approximately 35% of urban trees would be destroyed by a widespread *A. glabripennis* outbreak, and the potential economic loss would exceed US $1 trillion [6]. In China, *A. glabripennis* was estimated to be a medium- or high-risk pest in the Chinese provinces of Tibet, Xinjiang, GanSu, Heilongjiang, Zhejiang, and Beijing [8]. To accurately detect and estimate its population density, a great deal of research has indicated that olfactory signals play an important role in the intra- and interspecific communication of *A. glabripennis* adults [6,7,9]. Several sex pheromones and host kairomones have been identified, and single pheromones, as well as a combination of both semiochemical types, have been used as tools for detection and monitoring in *A. glabripennis* adults in the field [10,11], but few beetles were captured by the dozens of traps at Cornuda, Italy [12], and Paddock wood, UK [13]. Xu and Teale [9] also suggested that semiochemical-based traps baited with a mixture of host kairomones (plant volatile organic compounds) and sex pheromones have still not reached operational efficacy for *A. glabripennis* adults in field bioassays. For example, in Harbin, China, 42 beetles were trapped by 90 flight intercept panel traps with two pheromone components over approximately 27 days [14]. The mean number of beetles per panel trap was only 0.7–7 per trap per week with a combination of aldehydes, linalool oxide, and host kairomones [15]. Therefore, more effective tools for surveying and managing *A. glabripennis* adults are urgently needed.

Semiochemical, light, and vibrational stimuli-based traps are the most commonly used methods to detect, monitor and manage agricultural and forest insect pests [5]. The compound eyes are the main organ through which insects receive visual signals from mates and hosts for their intra- and interspecific communication. The compound eyes of adult *A. glabripennis* are kidney-shaped, and each eye has approximately 1000 facets [16]. An experiment showed that the mating behavior of male *A. glabripennis* was activated by visual stimuli from the female body [17]. In a previous study, we found that both adult females and males use complex mixed signals, including color, shape, texture (visual cues), and volatile cues, to locate host plants [18]. In addition, we also found that the bark color of branches plays an important role in host plant location and recognition [16].

Color is a characteristic that is determined by differing qualities of light being reflected or emitted by them. In agroecological systems, light as an attractant trap is widely used to monitor insect pest population density, predict pest outbreaks, and manage pest populations, such as *Helicoverpa armigera* (Hübner, 1809) and *Mythimna separata* (Walker, 1865) (Lepidoptera: Noctuidae), but the target pest is often a crepuscular or nocturnal creature [5,19]. The foraging peak of female and male *A. glabripennis* was 21:30 at night, and the percentages of foraging at night were significantly higher than those during the day [20], suggesting that a light-trapping strategy may be used to monitor and manage the population density of *A. glabripennis*. The phototactic behavior of insects to light sources depends on the characteristics of the light source, including light wavelength and intensity, and on other conditions, such as rhythmicity with photoperiod, illumination exposure time, and sex [19,21]. However, it remains unclear which light wavelength is most attractive to adult *A. glabripennis* and how these factors influence phototactic behavior.

In the present study, we investigated the influence of illumination exposure time, sex, diel rhythm, light wavelength, and intensity on the phototactic behavioral response of female and male *A. glabripennis*. Beetles (Coleoptera) show a preference for UV, violet, and green wavelength spectra, with wavelengths ranging from 350–440 and 500–560 [19,22]. Therefore, we selected four wavelength lights, including UV (365 nm), violet (420 nm and 435 nm), and green (515 nm) lights, to determine the effect of exposure time and diel rhythm on the phototactic behavior responses of females and males, thus, providing an appropriate test condition for the next experiment. Then, the effect of light wavelengths on the phototactic behavior responses of females and males was measured to identify the optimal wavelength of the light attraction. Finally, the different light intensities of the most preferred light wavelength were measured to quantify the illumination intensity preference of adults. We hope these results will provide a valuable reference for monitoring and managing population density and predicting *A. glabripennis* outbreaks.

## 2. Materials and Methods

### 2.1. Insects

In early July and late August, wild adult *A. glabripennis* males and females were obtained from *Salix matsudana umbraculifera* Rehd, 1949 and *S. babylonica* Linnaeus, 1753 plants at Gaocheng in Shijiazhuang, Hebei Province, China. Adults were maintained in rearing cages (d = 24 cm; h = 22 cm) at densities of 6 (three females and three males) per cage. A nylon gauze net with 16–20-mesh nylon gauze was used on the cages to prevent insect escapes. Adults were fed with fresh *S. babylonica* branches (L = 10–15 cm) that were replaced every 2 days and maintained at 27 ± 1 °C, 60 ± 10% relative humidity (RH), and a natural light/dark cycle (approximately 14 h (light): 10 h (dark)] in the laboratory (GMT+8).

### 2.2. Experimental Apparatus and Light Sources

To determine the phototactic behavior of *A. glabripennis*, a light–dark alternative apparatus using an opaque black acrylic board was self-designed based on the biological and behavioral characteristics of the insect and a previous study (Figure 1) [23]. The apparatus consisted of two test chambers at right angles to each other and an activity chamber that provided adequate space for insect movement. The activity chamber dimensions were L = 30 cm × H = 30 cm × W = 30 cm, and those of the test chambers were L = 50 cm × H = 30 cm × W = 30 cm. The test chamber provided an activity space for beetles to respond to illumination with different wavelength light-emitting diode (LED) lights and the side near the light source area was called the light area (Figure 1(Ba)), and the other side was called the dark area (Figure 1(Bb)). The top side of the light and dark areas was made of transparent acrylic board for observing the behavior of beetles. Half of the top was fixed to maintain the steadiness of the chamber, and the other half (Figure 1(Bf)) was used with a moveable transparent acrylic board to facilitate the removal and observation of insects after the test. Holes 10 cm in diameter were drilled in the tops of the activity chamber to add test insects, and a transparent acrylic board was added to the hole to avoid insect escape when test insects were introduced activity chamber. The end of the light area was fitted with a transparent acrylic board to prevent insect escape and facilitate light transmission, and the end of the dark area was fitted with an opaque acrylic board to avoid the entry of light. Two opaque black acrylic baffle boards (Figure 1Be)) were set up at the test and activity chambers. When the baffle was pushed down, only the LED light resource was turned on in the light area (Figure 1(Ba)). When the baffle was pushed up, the LED light resource could spread in the light area (Figure 1(Ba)) and activity chamber (Figure 1(Bc)), and there was no light in the dark area (Figure 1(Bb)). The light areas within 30 cm were used as the phototaxis areas, and the number of test insects in the phototaxis areas was regarded as the positive trapping number of adults. The number of insects in the dark area was used as the negative trapping number of adults. The outer and inner chambers were made of opaque black acrylic board to avoid the effect of light reflection, except for the top side of the light and dark areas. The experimental chamber was placed in an air-conditioned room to maintain a constant temperature of 25 ± 1 °C and 65 ± 10% RH under a completely dark environment.

Fourteen monochromatic LED lamps (365 nm, 385 nm, 400 nm, 420 nm, 435 nm, 475 nm, 500 nm, 515 nm, 560 nm, 595 nm, 600 nm, 630 nm, and 660 nm) distributing ultraviolet (UV) and visible light, were used as light stimuli at the end of the light area in this study (Table 1). The characteristic parameters of monochromatic LED lights are listed in Table 1. All lamps were made by Shenzhen Xinhongxian Electronics Technology Co., Ltd. (Shenzhen, China). The power of each lamp was 20 W. A rheostat was connected to the LED lights to adjust the intensity of illumination. The illumination intensity of the LED lights was measured by a digital lux meter (Tes-1337 B) from Taishi Instrument and Equipment Co., Ltd. (Changsha, China). The illumination intensities were measured at the center of the activity chamber because the phototactic behavioral response of beetles was attracted by light passing through a transparent acrylic board rather than under direct light. The light intensity of each test light was adjusted and always maintained at 180 lx, except for the 365 nm, 420 nm, and 435 nm wavelength LEDs, because the highest light intensity of those lights through the transparent acrylic board at the center of the activity chamber only reached 24 lx, 144 lx, and 36 lx, respectively.

### 2.3. Behavioral Experiments

To determine the influence of exposure time, diel rhythm, light wavelength, and light intensity on the phototactic behavioral response of adult *A. glabripennis*, a series of dual-choice experiments were performed in a completely dark indoor environment. Different light sources were installed at the end of the light area, while no light sources were installed at the end of the dark area. The 10 adult females or males, as one replicate in each test, were placed in the activity chamber through the insect entrance hole (Figure 1B,d) to adapt to the dark environment for 2 h. The LED lamps were turned on after 2 h, and the timer was started after the baffles were removed from the junction between the activity and the test chambers. The beetles either crawled or flew into the light or dark area. The number of beetles was recorded in the light area, dark area, and release area. The chamber was wiped with hexane and air-dried between trials to avoid residual adult volatile cues.

#### 2.3.1. Effect of Exposure Time on the Phototactic Behavioral Responses

We selected 4 wavelength spectra, including UV (365 nm), violet (420 nm and 435 nm), and green (515 nm), to analyze the effect of exposure time on the behavioral responses of adults in the time slot 19:00–21:00, providing a suitable exposure time for the next experiment. No LED lights were installed at the end of the dark area, while 4 different wavelength LED lights were installed at the end of the light area. During the observation period, the number of beetles in the light area, dark area, and release area were recorded at 15 min, 30 min, 45 min, 60 min, 75 min, 90 min, 105 min, and 120 min, respectively. Six-nine replicates (female: 9, 6, 7, and 8; male: 9, 7, 8, and 7) were measured at 365 nm, 420 nm, 435 nm, and 515 nm.

#### 2.3.2. Effect of Diel Rhythm on the Phototactic Behavioral Responses

The same wavelength spectra (365 nm, 420 nm, 435 nm, and 515 nm) were selected to analyze the effect of diel rhythm on the behavioral responses of adults. Experiments were conducted in five different time slots (9:00–11:00, 14:00–16:00, 19:00–21:00, 0:00–2:00, and 5:00–7:00). Based on the result of “Effect of exposure time on the phototactic behavioral responses” experiment, the number of beetles in the different areas was recorded at an exposure time of 120 min. Five to ten replicates [365 nm: 8, 9, 9, 6, and 6 (female) and 9, 9, 9, 5, and 5 (male); 420 nm: 6, 6, 6, 5, and 5 (female) and 7, 7, 7, 5, and 5 (male); 435 nm: 6, 8, 7, 5, and 5 (female) and 10, 8, 8, 5, and 5 (male); 515 nm: 9, 9, 8, 5, and 6 (female) and 7, 7, 7, 6, and 5 (male)] per time slot were measured for each light treatment.

#### 2.3.3. Monochromatic LED Light Preference

To determine the effect of different wavelengths on the phototactic behavioral responses of female and male *A. glabripennis*, a series of experiments were conducted. Based on the results of the experiment “Effect of diel rhythm on phototactic behavioral responses,” the test time was distributed in 5 different time slots (9:00–11:00, 14:00–16:00, 19:00–21:00, 0:00–2:00, and 5:00–7:00). The beetle either crawled or flew into the light or dark area. The number of beetles in the light area, dark area, and release area was recorded after an illumination time of 120 min. The samples of females and males were 5–10 replicates per time slot, and twenty-eight to thirty-nine replicates at five-time slots (female: 38, 29, 29, 28, 31, 29, 29, 33, 37, 32, 32, 33, 31, and 31; male: 37, 38, 31, 31, 36, 34, 29, 31, 32, 36, 34, 33, 39, and 35) were measured for 14 wavelengths of light (365–660 nm).

#### 2.3.4. Effect of Light Intensities on the Phototactic Behavioral Responses

The results of the “monochromatic LED light preference” tests showed that more female and male *A. glabripennis* were attracted by 420 nm and 435 nm lights than by the other monochromatic LED lights (Figure 3). The second peaks were at 365 nm, and the third peaks were at 560 nm and 595 nm lights. The illumination with a 365 nm lamp was excluded because the highest illumination intensity under the 365 nm lamp was 24 lx in the center of the activity chamber when passing through a transparent acrylic board to hardly divide the intensity in more detail. In addition, there was no significant difference in behavioral responses between females and males at wavelengths of 420 nm, 435 nm, 560 nm, and 595 nm. Females play an important role in lifestyle relative to males because females are required not only to search for feeding sites but also to find suitable ovipositing sites. Therefore, 420 nm, 435 nm, 560 nm, and 595 nm were selected to determine the effect of different intensities on the phototactic behavioral responses of female *A. glabripennis*. No LED lights were installed at the end of the dark area, while the most sensitive wavelength LED lights with different illumination intensities were installed at the end of the light area, including 9, 18, 36, 72, and 144 lx at 420 nm; 9, 18, and 36 lx at 435 nm; 45, 90, 180, 360, and 720 lx at 560 and 595 nm LED irradiation, respectively. During the observation period, the number of beetles in the different areas was recorded at 15 min, 30 min, 45 min, 60 min, 75 min, 90 min, 105 min, and 120 min. Six replicates per light intensity were measured for each light wavelength at 19:00–21:00.

### 2.4. Statistical Analysis

In this study, the trapping rate was used as an important parameter to evaluate the trapping level of different light sources for *A. glabripennis* at different treatments. The trapping rate was calculated using the following formula:(1)$$Trapping\;rate(\%)=\frac{Number\;of\;beetles\;in\;light\;area}{Total\;number\;of\;test\;beetles}\times 100$$

The significant differences in the trapping rates were evaluated using the Kruskal–Wallis H test between different exposure times. The significant differences in the trapping rates between different time slots under illumination with the same light source and between different light sources were analyzed using a generalized linear model (GLM) with Poisson distribution and a log link function followed by the Bonferroni test. The trapping rate between females and males was compared for each light source with a Mann–Whitney *U* test. Under illumination with wavelengths of 420 nm, 435 nm, 560 nm, and 595 nm, the effect of different intensities on the phototactic response rate of females at different exposure times was analyzed using a nonparametric Kruskal–Wallis *H* test. Nonresponders were recorded but excluded from the analysis. All experimental data were statistically analyzed using SPSS Statistics v. 21.0 (IBM Corp., Armonk, NY, USA) for Windows.

## 3. Results

### 3.1. Effect of Exposure Time on the Phototactic Behavioral Response

As shown in Figure 2, generally, the trapping rate of females and males increased gradually when the exposure time was prolonged. Both females and males showed no significant difference in the trapping rate between different exposure times at wavelengths of 365 nm, 420 nm, 435 nm, and 515 nm LED lights (Figure 2), but the trapping rate of adults at an exposure time of 15 min was lower than that at the other exposure times. For example, the trapping rate of females and males was 55.55% and 55.55% at 365 nm at 15 min exposure time and 61.11% and 64.44% at 120 min, respectively. The trapping rate of males is 57.15% at 420 nm at 15 min exposure time and 72.86% at 120 min. The trapping rate of females and males was 68.75% and 48.75% at 15 min exposure time at 435 nm and 74.29% and 60% at 120 min, respectively. The trapping rate of females and males was 37.50% and 52.85% at 15 min exposure time at 515 nm and 47.50% and 55.71% at 120 min, respectively (Figure 2). The trapping rate also showed no significant difference between females and males at the different exposure times under different wavelengths (365 nm: 25.000 < *U* < 39.500, 0.190 < *p* < 0.931; 420 nm: 14.000 < *U* < 19.500, 0.366 < *p* < 0.836; 435 nm: 10.000 < *U* < 24.000, 0.073 < *p* < 1.000; 515 nm: 13.000 <*U* < 26.000, 0.094 < *p* < 0.867).

### 3.2. Effect of Diel Rhythm on the Phototactic Behavioral Response

The behavioral responses of both females and males were analyzed at five different time slots to compare the effect of diel rhythm on the attraction of LED lights (Figure 3). The results showed that the behavioral responses of adults fluctuated with changes in diel rhythm. Under 420 nm and 435 nm illumination, both females and males exhibited higher preference at night (0:00–2:00) than at other time slots, and the trapping rates of females and males were 74% and 82% (420 nm) and 78% and 76% (435 nm), respectively (Figure 3B,C; 420 nm, female: *X*^2^ = 2.841, *df* = 4, *p* = 0.585, male: *X*^2^ = 18.283, *df* = 4, *p* = 0.001; 435 nm, female: *X*^2^ = 13.719, *df* = 4, *p* = 0.003, male: *X*^2^ = 17.010, *df* = 4, *p* = 0.002). At 365 nm and 515 nm, females showed higher phototactic responses at 5:00–7:00 (Figure 3A,D; trapping rate: 66.67% at 365 nm, *X*^2^ = 2.624, *df* = 4, *p* = 0.623; 61.67% at 515 nm, *X*^2^ = 21.826, *df* = 4, *p* < 0.001), while males showed higher phototactic responses at 9:00–11:00 (Figure 3A,D; trapping rate: 71.11% at 365 nm, *X*^2^ = 28.229, *df* = 4, *p* < 0.001; 67.14% at 515 nm, *X*^2^ = 27.801, *df* = 4, *p* < 0.001).

### 3.3. Monochromatic LED Light Preference

The phototactic behavioral responses of both females and males showed a significant difference among 14 different wavelength LED cues (Figure 4B,D; female: *X*^2^ = 767.792, *df* = 13, *p* < 0.001; male: *X*^2^ = 385.236, *df* = 13, *p* < 0.001). Females and males exhibited similar preferences under 14 different LED lamps, and both females and males showed the highest preference for 420 nm and 435 nm LED lights, which are violet light ranges. The trapping rates of females were 70.71% and 70% under 420 and 435 nm LED lights, respectively (Figure 4B), while the trapping rates of males were 70.32% and 68.61% under 420 and 435 nm LED lamps, respectively (Figure 4D). In addition, both females and males also preferred LED lights at 365 nm (UV region), 455 nm and 475 nm (blue region), 560 nm (green region), and 595 nm (yellow region). The lowest trapping rate was in the red region (660 nm, 39–43%).

Moreover, the trapping rates of females showed a significantly higher for 400 nm, 435 nm, and 515 nm at nighttime (19:00–7:00) than in the daytime (9:00–16:00) (Figure 4A, 400 nm: *X*^2^ = 29.758, *df* = 4, *p* < 0.001; 435 nm: *X*^2^ = 15.719, *df* = 4, *p* = 0.003; 515 nm: *X*^2^ = 21.826, *df* = 4, *p* < 0.001), while the trapping rates of males showed a significantly higher for 400 nm, 420 nm, 435 nm, and 560 nm at nighttime (19:00–7:00) than in the daytime (9:00–16:00) (Figure 4C, 400 nm: *X*^2^ = 32.775, *df* = 4, *p* < 0.001; 420 nm: *X*^2^ = 18.283, *df* = 4, *p* = 0.001;435 nm: *X*^2^ = 17.010, *df* = 4, *p* = 0.002; 560 nm: *X*^2^ = 23.202, *df* = 4, *p* < 0.001). When adult insects were given light cues at 500 nm and 630 nm, the trapping rate of males was higher than that of females (Table 2). There was no significant difference in the trapping rate between females and males under illumination with other wavelengths (Table 2).

### 3.4. Effect of Light Intensities on the Phototactic Behavioral Response

The behavioral responses of females were determined at different light intensities at 420 nm, 435 nm, 560 nm, and 595 nm, which were the wavelengths most preferred by females and males (Figure 5). Overall, 18 lx was the most attractive for females under short-wavelength illumination (420 nm and 435 nm), while 180 lx was the most attractive under long-wavelength illumination (560 nm and 595 nm), but there was no significant difference between the different light intensities at a 120 min exposure time. Under 420 nm illumination, the trapping rate of females at 36 lx was lower than the other four light intensities after light exposure times of 15, 30, 45, and 90 min (15 min: *X*^2^ = 13.529, *p* = 0.009; 30 min: *X*^2^ = 9.694, *p* = 0.046; 45 min: *X*^2^ = 11.549, *p* = 0.021; 90 min: *X*^2^ = 9.578, *p* = 0.048; Figure 5A). Female adults were more attracted to 90 lx and 180 lx under 560 nm illumination than to 45 lx, 360 lx, and 720 lx after light exposure times of 15 min and 30 min (15 min: *X*^2^ = 10.515, *p* = 0.033; 30 min: *X*^2^ = 10.354, *p* = 0.035; Figure 5C). There was no significant difference between the different light intensities at 435 nm (Figure 5B: 0.544 < *X*^2^ < 3.613, 0.164 < *p* < 0.762) and 595 nm (Figure 5D: 2.028 < *X*^2^ < 5.965, 0.731 < *p* < 0.202).

## 4. Discussion

Many insects show positive phototaxis to artificial lights, and this characteristic is currently used to detect, monitor and manage insect pests in agroecology systems, especially nocturnal insects [19]. Light traps have been used to monitor the population density of beetles, including Tenebrionidae, Curculionidae, Pselaphidae, Silvanidae, Cerambycidae, and Scolytinae [24,25,26]. Red LED lamps are more attractive to *Tribolium castaneum* (Herbst), *Sitophilus zeamais* (Motschulsky), *Lasioderma serricorne* (Fabricius), and Scolytinae, while blue LED lamps are more attractive to *Sitophilus oryzae* (Linnaeus) [25,26]. For jewel beetles, flower-visiting preferred yellow traps over green, and non-flower-visiting tended to green traps [27]. For longhorn beetles, flowering-visiting species were caught by flower-related colors, such as yellow, green, and blue, while non-flower-visiting were more attracted by long wavelength colors, i.e., red and brown [28]. However, in the present study, we show that the phototactic behavioral responses of female and male *A. glabripennis* were influenced by diel rhythm, sex, and light wavelength and intensity, while illumination exposure time did not have significant effects. Both female and male *A. glabripennis* have four spectral sensitivity regions to illumination with 365–660 nm LEDs, including violet (420, 435 nm), ultraviolet (365 nm), blue (455 nm and 475 nm), and green (560, 595 nm) regions (Figure 4). The difference in color and spectrum preference may be due to beetles’ different dietary and environmental habits. For example, flower-visiting beetles feed on nectar on yellow, green, or blue flowers, Scolytinae beetles mainly feed on the main brown trunk, while adults *A. glabripennis* mainly feed on branch barks with a diameter of 2–4 cm.

Many insect photoreceptor cells contain three types of visual pigments that are sensitive to UV (UV-opsin), blue (SW-opsin), and green (LW-opsin) wavelength regions, such as those of *Nephotettix cincticeps* [29], *Plutella xylostella* [30], *Spodoptera exigua* [21], and *Diaphorina citri* [31]. Therefore, they exhibit a highly sensitive behavioral response to light sources in the UV, blue, and green wavelength regions. However, it has been proposed that the SW opsin class (blue-sensitive opsin) has been lost based on molecular evidence in beetle lineages, such as Lampyridae [32], Buprestidae [33], *Tribolium castaneum* [34], and *A. glabripennis* [35]. However, many beetles exhibit a preference for blue color or blue wavelength lights, especially longhorned beetles. A color-based trapping test showed that the number of longhorned beetle species caught in yellow and blue traps was significantly higher than that caught in black traps [28]. Adult *Arhopalus ferus* (Mulsant, 1839) (Coleoptera: Cerambycidae) are also significantly more attracted to UV-blacklight (BL) and UV-blacklight blue (BLB) than to yellow light traps [36]. In this study, the results showed that female and male *A. glabripennis* exhibited significant preferences for the blue light region (Figure 4). The phototactic behavioral response rate of females and males was 59.66%–66.21% in the blue region with wavelengths of 455 nm and 475 nm (Figure 4). This result suggests that both females and males exhibit a strong preference for the blue light region. Sharkey et al. [35] suggest that UV duplications and subsequent amino acid changes may lead to an extension of the light signal recognition capacity into short and blue light wavelengths to overcome the loss of SW-opsin in Coleoptera. However, whether the stronger phototactic response to blue wavelengths in *A. glabripennis* is mediated by SW-opsins or UV-opsins requires further research to determine the types of opsins.

Light as an attractant to trap and kill insect pests plays a crucial role in monitoring pest population density and forecasting pest outbreaks [19]. The sensitivities of target pests to light sources of different wavelengths are crucial for the successful application of light traps. In agroecology systems, the most sensitive wavelengths of many insect pests to artificial light sources have been identified, such as UV and green wavelength spectra, which were favored by Lepidoptera, while Coleoptera showed a preference for UV and violet wavelengths [19]. Furthermore, due to the natural light present in the daytime, the premise of light traps to monitor and manage insect pests is that the target insect pests are nocturnal or crepuscular [5]. In a previous study, we found that the mating and foraging behaviors of adult *A. glabripennis* occur throughout the day, with foraging behavior peaking at night (21:30) [20]. In addition, some kinds of Cerambycida, which include *Elaphidion lanatum* Chevrolat, *Callipogon barbiflavum* Chevrolat, and *Stenodontes chevrolati* Gahan, were captured by mercury vapor lamp, UV light trap, or fluorescent lighting [37,38]. Therefore, light trapping may improve the monitoring and management strategies of *A. glabripennis*.

Many management strategies have been widely used to monitor and control *A. glabripennis*, including the eradication of infested trees [12], chemical insecticide treatment [39,40], and biological control of natural enemies [41,42,43]. However, eradication programs have proven to be one of the most effective methods. More than 45% of eradication campaigns were successful in 2009–2019 [6]. Accurate outpost warning of *A. glabripennis* adult or identification of infestation symptoms plays an important role in the successful eradication campaign [12], but semiochemical-based (sex pheromones and host kairomones) traps data have shown that the number of adult *A. glabripennis* trapped is still limited in the field [44]. For example, only two *A. glabripennis* females were caught in Italy [12], and no beetles were captured in England [13] during the eradication program. In the field, the trapping data showed that a number of beetles (Scolytinae) were captured by combining ethanol-baited traps with green (525 nm) and UV (395 nm) LED lights more than ethanol-baited traps [45]. In addition, we found that both females and males used visual and chemical cues to locate and recognize host plants in previous studies [16,18]. Therefore, comparing the catch capacity of light traps, semiochemicals, and their integration for adult *A. glabripennis* needs to be further investigated in the field, thus, providing effective tools for monitoring and management of this beetle.

The endogenous circadian clock influences almost every aspect of the behavior and physiology of humans, animals, insects, some bacteria, and even plant rhizosphere microbial communities, including physiological functions, mating behaviors, metabolic regulation, and hormone secretion [46,47,48,49]. The phototactic behavioral response rates of both females and males displayed circadian rhythms under illumination in the UV, violet, and blue wavelength regions (Figure 3 and Figure 4). Opsins are the molecular basis of insect vision, which combine with chromophores to absorb light signals and then detect host plants and mates [50]. The fluctuation in the phototactic rate may indicate a change in the relative expression quantity of opsin in adult *A. glabripennis*, similar to that of *Spodoptera exigua* (Hübner, 1808) [21]. The three opsin genes (se-uv, bl, and lw) of *S. exigua* exhibited circadian rhythms in relative expression quantity and showed higher expression levels during daytime than nighttime, except for se-bl in females [21]. Whether the change in relative expression also exhibited circadian rhythms in the opsin of *A. glabripennis* needs further research.

To improve the effectiveness of light traps, a light wavelength that is the most sensitive to target insect pests must be identified [51]. Under wavelengths of 420 nm and 435 nm illumination, both females and males displayed higher phototactic responses during the nighttime than during the daytime (Figure 3). Therefore, we suggest that 420 nm and 435 nm LED illumination can be selected to monitor and detect the population density of *A. glabripennis*. When light traps are used to detect and control insect pests, they can also influence the natural enemies of the pests [52]. Therefore, the phototactic behavioral responses of natural enemies should be considered in the monitoring and detection of insect pests. The ectoparasitoid beetle *Dastarcus helophoroides* (Fairmaire, 1881) (Coleoptera: Bothrideridae) is an important natural enemy of wood-boring insects and is widely used in controlling cerambycid beetles, including *A. glabripennis* and *Monochamus alternatus* Hope, 1842 (Coleoptera: Cerambycidae) [53]. The photopositive behavioral response of *D. helophoroides* was significantly influenced by light intensity, and a 700 nm wavelength with 7 lx light intensity was most attractive to adults among the test intensities, but there was no significant difference in the phototactic response rates between red, green, and violet region wavelengths, while the phototactic response rate decreased significantly when the illumination intensities were more than 10 lx [54]. In the present study, we found that female *A. glabripennis* did not exhibit differences among the different light intensities at wavelengths of 420 nm and 435 nm at a 120 min exposure time (Figure 5). Therefore, light intensity should be enhanced to reduce the trapping of *D. helophoroides* at wavelengths of violet light (420 nm and 435 nm), for instance, illumination with 180 lx.

The phototactic behavioral responses of insects are also influenced by light exposure time [19]. We found that the phototactic rate of adults tended to be enhanced as the light exposure time was prolonged, but there was no significant difference between different light exposure times (Figure 2), indicating that the movement of screening pigment granules is relatively fast in the compound eyes of *A. glabripennis* from dark conditions to light conditions. The pigment is contracted peripherally of the crystalline cones in dark-adapted eyes, whereas in light-adapted eyes, the pigment diffuses to adapt to changes in the light environment [55]. The rapid movement and distribution of the screening pigment can increase the sensitivity of insects to light, improving the adaptation ability of insects to the environment [19]. Structure research on compound eyes shows that adult *M. alternatus* have extremely high absolute sensitivity to light [56]. Therefore, we inferred that *A. glabripennis* and *M. alternatus* locate suitable host plants by increasing their sensitivity to the spectrum.

Each insect pest uses multiple sense signals for intra- and interspecies communication, such as light, odor, color, sound, and vibration, so multiple channels must be considered in the monitoring and management of insect pests [5]. For instance, color contrast between ripening fruits and senesced foliage was used by female *Drosophila suzukii* (Matsumura, 1931) (Diptera: Drosophilidae) to locate suitable hosts. Therefore, red, which mimics the color of ripe fruits, was used to decorate the surface of the trapping device, combined with food attractants to monitor and manage *D. suzukii* [57]. The vibrational signal of female *Halyomorpha halys* (Stal, 1855) (Heteroptera, Pentatomidae) can improve the performance of pheromone traps in controlled conditions, and in the field, therefore, a minishaker that plays the female’s call was combined with the aggregation pheromone to overcome sex selection [58]. A previous study showed that forest green color could enhance the behavioral response of female and male *A. glabripennis* to chemical signals from the cut branches of the host plant *A. negundo* [16]. In this study, both female and male *A. glabripennis* exhibited a high phototactic behavioral response to LED sources in the UV, violet, blue, and green wavelength regions. Moreover, both females and males displayed higher phototactic responses at night under illumination at 420 nm and 435 nm wavelengths. Therefore, integrated light (420 nm and 435 nm), color, and pheromone signals should be considered to develop an innovative multimodal trapping system for *A. glabripennis*.

## Figures and Tables

**Figure 1 insects-14-00465-f001:**
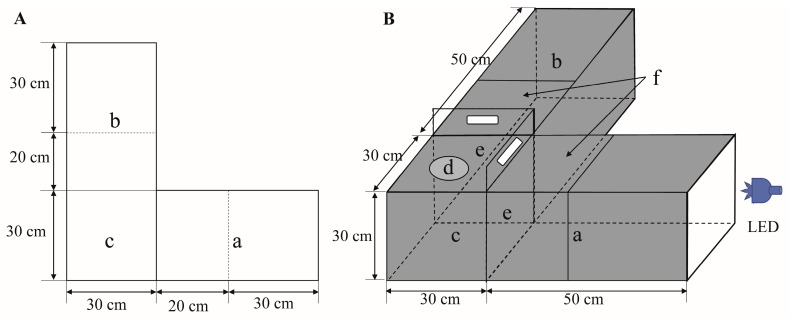
Planar (**A**) and stereoscopic (**B**) schematics of the behavioral bioassay chamber used to test the attractiveness of LED light cues in female and male *A. glabripennis*. Storage chambers (**a**,**b**); activity chamber (**c**); test insect release point (**d**); black opaque acrylic boards (**e**) and moveable transparent acrylic board (**f**). LED: light emitting diode.

**Figure 2 insects-14-00465-f002:**
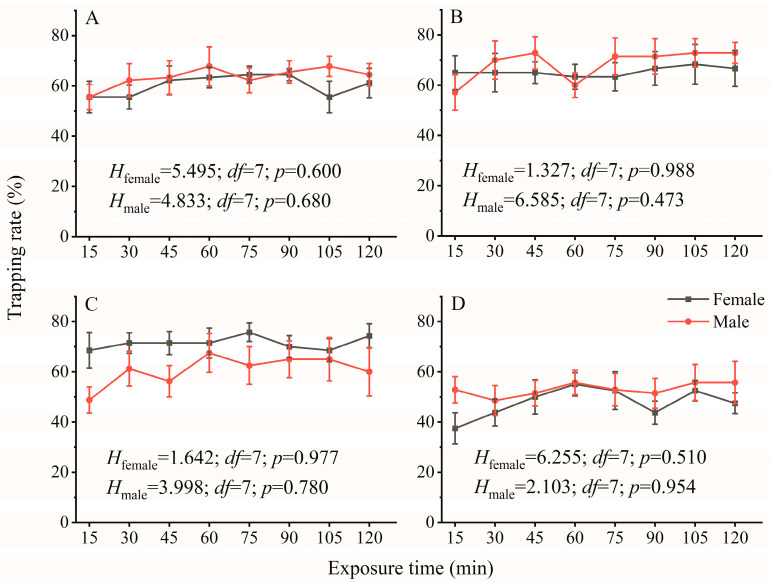
The effect of exposure time on the phototactic behavioral response rate of females and males at wavelengths of 365 nm (**A**), 420 nm (**B**), 435 nm (**C**), and 515 nm (**D**). The significant differences in phototactic behavioral responses to LED lights among different exposure times were analyzed using Kruskal–Wallis H test; α = 0.05. The samples of females and males were 9, 6, 7, and 8 replicates and 9, 7, 8, and 7 replicates at 365 nm, 420 nm, 435 nm, and 515 nm, respectively. The data are presented as the mean ± SE.

**Figure 3 insects-14-00465-f003:**
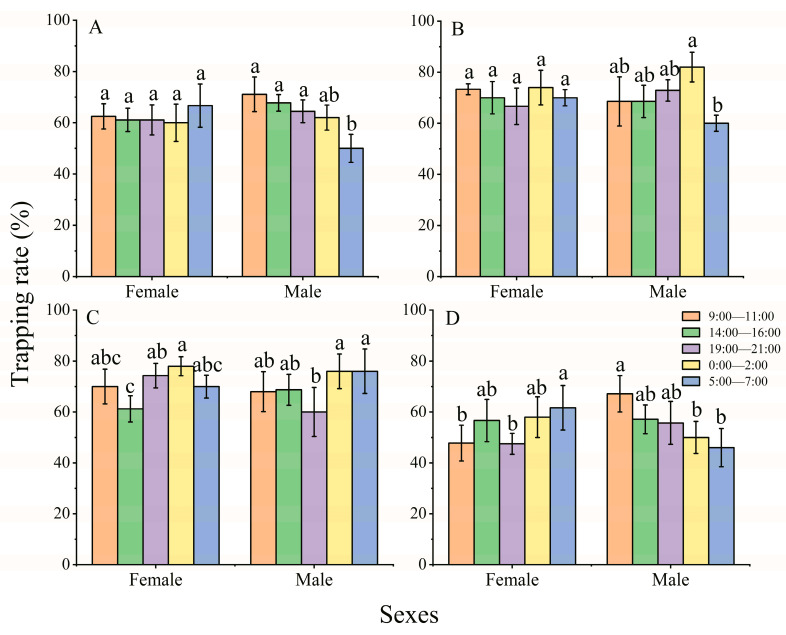
The effect of diel rhythm on the phototactic response rate of females and males at wavelengths of 365 nm (**A**), 420 nm (**B**), 435 nm (**C**), and 515 nm (**D**). The same lowercase letters on top of the bar indicate no significant difference among the trapping rate at different time slots under 365 nm, 420 nm, 435 nm, and 515 nm wavelength, respectively (GLM with Poisson distribution and a log link function followed by Bonferroni test; α = 0.05). The behavioral response of adults to four wavelengths of light at five time slots [*n* = 8, 9, 9, 6, and 6 (female) and 9, 9, 9, 5, and 5 (male) at 365 nm; 6, 6, 6, 5, and 5 (female) and 7, 7, 7, 5, and 5 (male) at 420 nm; 6, 8, 7, 5, and 5 (female) and 10, 8, 8, 5, and 5 (male) at 435 nm; 9, 9, 8, 5, and 6 (female) and 7, 7, 7, 6, and 5 (male) at 515 nm]. The data are presented as the mean ± SE.

**Figure 4 insects-14-00465-f004:**
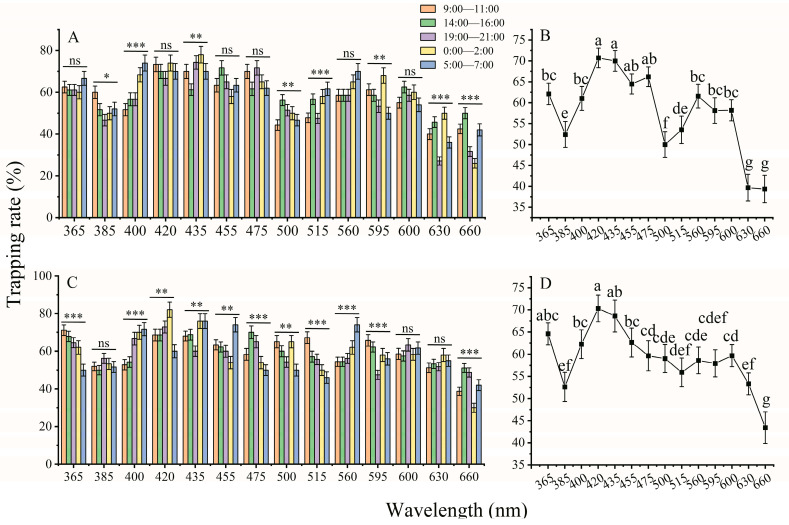
Phototactic behavioral response rates of female (**A**,**B**) and male (**C**,**D**) *A. glabripennis* to different wavelengths of light under five time slots. The same lowercase letters on top of the line indicate no significant difference among the phototactic behavior response rates under the different wavelengths (GLM with Poisson distribution and a log link function followed by Bonferroni test; α = 0.05). ns: *p* > 0.05, *: *p* < 0.05, **: *p* < 0.01, ***: *p* < 0.001. The samples of females and males were 5–10 replicates per time slot, and the total samples of females and males were 38, 29, 29, 28, 31, 29, 29, 33, 37, 32, 32, 33, 31, and 31 replicates and 37, 38, 31, 31, 36, 34, 29, 31, 32, 36, 34, 33, 39, and 35 replicates at 365–660 nm, respectively. The data are presented as the mean ± SE.

**Figure 5 insects-14-00465-f005:**
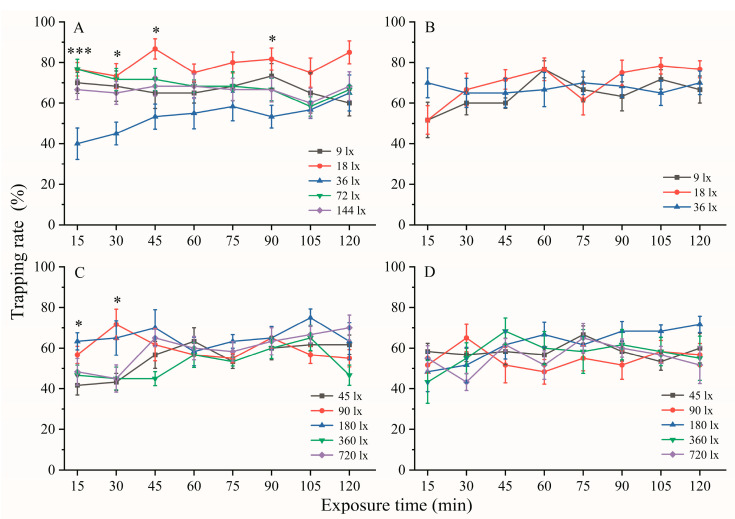
Positive phototactic response rate of female *A. glabripennis* at wavelengths of 420 nm (**A**), 435 nm (**B**), 560 nm (**C**), and 595 nm (**D**) after different light exposure times. *: *p* < 0.05, ***: *p* < 0.001 indicate that differences in phototactic behavioral response rates among the different light intensities were assessed with Kruskal–Wallis *H* test at the same exposure time under the same wavelength lights. There are six replicates per light intensity. The data are presented as the mean ± SE.

**Table 1 insects-14-00465-t001:** The parameter of LED lights and samples in the behavioral response experiments.

Light Wavelength (nm)	Spectrum Region	Main Wavelength (nm)	Peaking of Wavelength (nm)	Band Width of Peaking (nm)	Luminance (Lux) #
365	Ultraviolet	472.0	365.6	12.7	24
385	Ultraviolet	388.0	385.2	12.0	180
400	Violet	370.2	402.1	15.8	180
420	Violet	427.1	420.9	17.6	144
435	Violet	441.0	435.0	17.2	36
455	Blue	460.3	455.5	20.9	180
475	Blue	477.2	473.9	25.4	180
500	Green	502.8	499.2	26.3	180
515	Green	522.4	515.7	32.0	180
560	Green	568.1	562.1	46.9	180
595	Yellow	592.6	596.3	14.6	180
600	Orange	595.2	599.5	15.7	180
630	Red	620.3	631.6	18.6	180
660	Red	641.1	659.6	17.1	180

#: luminance intensity indicates the light intensity in the central site of the activity chamber through a transparent acrylic board.

**Table 2 insects-14-00465-t002:** Comparison of phototactic behavioral response rate of females with that of males under 14 different wavelengths.

Wavelength (nm)	Sample (F, M)	Statistical Value	*p*
365	38, 37	627.000	0.410
385	29, 38	524.500	0.734
400	29, 31	401.500	0.470
420	28, 31	431.000	0.963
435	31, 36	520.000	0.626
455	29, 34	470.000	0.746
475	29, 29	324.500	0.125
500	33, 31	364.500	**0.045**
515	37, 32	559.500	0.691
560	32, 36	524.500	0.518
595	32, 34	538.500	0.943
600	33, 33	503.500	0.590
630	31, 39	346.000	**0.002**
660	31, 35	490.000	0.494

Note: Differences in phototactic behavioral response rate between females and males for 14 different wavelengths were assessed using the Mann–Whitney *U* test. α = 0.05; significant values are in bold type.

## Data Availability

The data presented in this study are available on request from the corresponding author.

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
