# Peer review of "Research on Photoinduction-Based Technology for Trapping Asian Longhorned Beetle (Anoplophora glabripennis (Motschulsky, 1853) (Coleoptera: Cerambycidae)"

_insects, 2023, doi:10.3390/insects14050465_

Round 1

Reviewer 1 Report

The authors describe a new potential method for trapping the dangerous invasive long-horned beetle Anoplophora glabripennis. After major revisions I recommend it for publication.

The whole article is based only on laboratory tests, it is important to test this method in the field. In case of functionality it can be innovative. Most of the comparisons in the discussion are based on butterflies. The discussion needs to be reworked so that detailed comparisons are made with beetles. Whether it is trap color, light trapping, or adding volatiles to light traps. There is plenty of literature, e.g.

Frost S.W. 1964: Insects taken in light traps at the Archbold Biological Station, Highlands County, Florida. The Florida Entomologist 47: 129-161.

Gorzlancyk A.M., Held D.W., Kim D.-J., Ranger C.M. 2013: Capture of Xylosandrus crassiusculus and other Scolytinae (Coleoptera: Curculionidae) in response to visual and volatile cues. Florida Entomologist 96: 1097–1101.

Marchioro M., Battisti A., Faccoli M. 2020: Light traps in shipping containers: A new tool for the early detection of insect alien species. Journal of Economic Entomology 113: 1718–1724.

Werle C.T., Bray A.M., Oliver J.B., Blythe E.K., Sampson B.J. 2014: Ambrosia beetle (Coleoptera: Curculionidae: Scolytinae) captures using colored traps in southeast Tennessee and south Mississippi. Journal of Entomological Science 49: 373-382.

Thomas M.C., Turnbow R.H. 2007: Cerambycidae new to Andros Island, Bahamas (Coleoptera). The Coleopterists Bulletin 61: 581-588.

Lingafelter S.W., Horner N.V. 1993: The Cerambycidae of north-central Texas. The Coleopterists Bulletin 47: 159-191.

L42 – word „control“ must be replaced by the word „protection“

L64 – (2021) must be replaced by [9], its citation in references

L100-111 – „In addition…..“ this paragraph belongs in the results or discussion

L196-197 – The first sentence belongs in the introduction

Figure 3 - the lowercase letters in the picture are vaguely explained. Please provide a more accurate description

L343 - error in table name, it should be Table 2

L346 - significant values are not highlighted in the table

L393 – (2017) must be replaced by [30], its citation in references

L417-423 – „At present….host plants [16,18]“ this paragraph is just a statement from the introduction, it should be deleted or significantly revised

L445 – „A.glabripennis“ must be written in italics

Author Response

May 11, 2023

Insects, section Insect Pest and Vector Management; Special Issue: Managing Invasive Insects: Good Intentions, Hard Realities

Dear Dr. Ivan and reviewers

We are very grateful for the helpful comments from subject editor and reviewers according to our manuscript (Manuscript ID insects-2375491: Research on photoinduction-based technology for trapping Asian longhorned beetle). We also thank you for your invaluable help during the assessment of our manuscript.

Thank you for considering this work to publish in the Insects.

We have revised carefully our manuscript based on the comments, please see the followed details.

Sincerely,

Fei Lyu

College of Forestry

Hebei Agricultural University

Baoding, Hebei, China

Response to Reviewers

*************************

Reviewer 1

The authors describe a new potential method for trapping the dangerous invasive long-horned beetle Anoplophora glabripennis. After major revisions I recommend it for publication.

The whole article is based only on laboratory tests, it is important to test this method in the field. In case of functionality it can be innovative. Most of the comparisons in the discussion are based on butterflies. The discussion needs to be reworked so that detailed comparisons are made with beetles. Whether it is trap color, light trapping, or adding volatiles to light traps. There is plenty of literature, e.g.

Frost S.W. 1964: Insects taken in light traps at the Archbold Biological Station, Highlands County, Florida. The Florida Entomologist 47: 129-161.

Gorzlancyk A.M., Held D.W., Kim D.-J., Ranger C.M. 2013: Capture of Xylosandrus crassiusculus and other Scolytinae (Coleoptera: Curculionidae) in response to visual and volatile cues. Florida Entomologist 96: 1097–1101.

Marchioro M., Battisti A., Faccoli M. 2020: Light traps in shipping containers: A new tool for the early detection of insect alien species. Journal of Economic Entomology 113: 1718–1724.

Werle C.T., Bray A.M., Oliver J.B., Blythe E.K., Sampson B.J. 2014: Ambrosia beetle (Coleoptera: Curculionidae: Scolytinae) captures using colored traps in southeast Tennessee and south Mississippi. Journal of Entomological Science 49: 373-382.

Thomas M.C., Turnbow R.H. 2007: Cerambycidae new to Andros Island, Bahamas (Coleoptera). The Coleopterists Bulletin 61: 581-588.

Lingafelter S.W., Horner N.V. 1993: The Cerambycidae of north-central Texas. The Coleopterists Bulletin 47: 159-191.

Response: done

Thank you for your comments.

We have revised discussion sections to increase the content of detailed comparisons with beetles as suggested. (Lines 376-383, 390-393, pages 12-13; lines 426-428, page 13; lines 436-442, page 13 in the marked-up version)

L42 – word „control“ must be replaced by the word „protection“

Response: done

Thank you for your comments.

We have revised as suggested. (Line 43, page 1 in the marked-up version)

L64 – (2021) must be replaced by [9], its citation in references

Response: done

Thank you for your comments.

We have revised as suggested. (Line 65, page 2 in the marked-up version)

L100-111 – „In addition…..“ this paragraph belongs in the results or discussion

Response: done

Thank you for your comments.

We have deleted as suggested. (Lines 101-103, page 3 in the marked-up version)

L196-197 – The first sentence belongs in the introduction

Response: done

Thank you for your comments.

We have deleted the first sentence in this paragraph as suggested. (Lines 194-195, page 6 in the marked-up version)

Figure 3 - the lowercase letters in the picture are vaguely explained. Please provide a more accurate description

Response: done

Thank you for your comments.

We have revised as suggested. (Lines 302-303, page 9 in the marked-up version)

L343 - error in table name, it should be Table 2

Response: done

Thank you for your comments.

We have revised the table name in the manuscript as suggested. (Line 343, page 10 in the marked-up version)

L346 - significant values are not highlighted in the table

Response: done

Thank you for your comments.

We have highlighted the significant value in the manuscript as suggested. (Line 344, page 11 in the marked-up version)

L393 – (2017) must be replaced by [30], its citation in references

Response: done

Thank you for your comments.

We have revised as suggested. (Line 411, page 13 in the marked-up version)

L417-423 – „At present….host plants [16,18]“ this paragraph is just a statement from the introduction, it should be deleted or significantly revised

Response: done

Thank you for your comments.

We have significantly revised as suggested. (Lines 436-442, page 13 in the marked-up version)

L445 – „A.glabripennis“ must be written in italics

Response: done

Thank you for your comments.

We have revised as suggested. (Line 471, page 14 in the marked-up version)

Reviewer 2

Dear authors,

In your paper is presented result of the light trap attraction study to collect Cerambycidae species adults, and it is very interesting to those who has experience in collecting invasive insects in forests. You are absolutely right when writing about difficulties to collect invasive species using traditional traps. Pheromone traps are working on narrow spectrum of species, light traps just occasionally attract flight beetles, canopy traps very hard to mount. That is why your test of different light wavelength lamp attraction to collect longhorn beetle species is actual at present time.

Several questions appeared after acquaintance with your work. 

— beetles living in nature adapted to natural source light, such as day light (380 – 780 nm) or fire (589 nm). Your lighting were 365 nm, 420 nm, 435 nm and 515 nm. Why did not used standard lamps too? Why spectrum of fire has not been tried? 

Response: done

Thank you for your comments.

In previous study, beetles (Coleoptera) show a preference for UV, violet and green wavelength spectra, with wavelengths ranging from 350-440 and 500-560 [1,2]. In addition, we also found that forest green can enhanced the response of adult ALB to odor cues from the cut branches of host plants [3]. Therefore, we selected 14 wavelength of LED lights about 20 nm intervals to determine the phototactic behavioral response rate of adults A. glabripennis in the experiment, arranging 365 nm-660 nm.

The standard lamps, 660 nm-780 nm LED and fire lamps (589 nm) will be test in the further.

  1. Van der Kooi, C.J.; Stavenga, D.G.; Arikawa, K.; Belusic, G.; Kelber, A. Evolution of insect color vision: from spectral sensitivity to visual ecology. Annu. Rev. Entomol. 2021, 66, 435-461, doi:10.1146/annurev-ento-061720-071644.
  2. Kim, K.; Huang, Q.; Lei, C. Advances in insect phototaxis and application to pest management: a review. Pest Manag. Sci. 2019, 75, 3135–3143, doi:10.1002/ps.5536.
  3. Lyu, F.; Hai, X.; Wang, Z. Green-colored paperboard enhances the Asian longhorned beetle response to host plant odor cues. J. Pest Sci. 2021, 94, 1345-1355, doi:10.1007/s10340-020-01318-3.

— what do you think why violet spectrum (420 nm and 435 nm) is more efficient to attract the beetle adults?

Response: done

Thank you for your comments.

In present study, the result of experiments showed that the trapping rate of females and males under the wavelength of 420 nm and 435 nm is higher than the other wavelengths in the 14 different wavelengths (see below figure). Therefore, we suggested that (420 nm and 435 nm) is more efficient to attract the beetle adults under the wavelength range of 365 nm - 660 nm.

— To compare catching efficiency of different trap types the control trapping should be held. For example, different type traps, such as: pheromone, canopy, flight-break etc. should have been applied and result compared with your light experiment 

Response: done

Thank you for your comments.

Indeed, different type traps, such as: pheromone, canopy, flight-break etc. have been applied to agricultural pests, for example, Drosophila suzukii, Halyomorpha halys, but there are still relatively few reports of combined signals in the monitoring and detection of adult A. glabripennis.

Therefore, the catch capacity of light traps, semiochemicals, and their integration for adult A. glabripennis will be compare in the further in the field to provide effective tools for monitoring and management of this beetle.

In addition, we have revised discussion sections to increase the content of detailed comparisons with beetles as suggested reviewer 1. (Lines 376-383, 390-393, pages 12-13; lines 426-428, page 13; lines 436-442, page 13 in the marked-up version)

All these questions are proposed to enhance your future study, the manuscript could be published at present version.

With the kindest wishes.

Reviewer 3

Recommended amendments and adjustments:

At the first mention of the Latin name, it is necessary to give the full name of the species with the author and year of description. I recommend doing this in the title of the article. 
Research on photoinduction-based technology for trapping Asian longhorned beetle (Anoplophora glabripennis (Motschulsky, 1853) (Coleoptera: Cerambycidae)

Response: done

Thank you for your comments.

We have revised as suggested. (Lines 2-4, page 1 in the marked-up version)

line 51: Anoplophora glabripennis delete (Motschulsky)

Response: done

Thank you for your comments.

We have revised as suggested. (Line 52, page 2 in the marked-up version)

line 56: delete adult, because the larva is the harmful stage.

Response: done

Thank you for your comments.

We have revised as suggested. (Line 57, page 2 in the marked-up version)

line 86: Helicoverpa armigera (Hübner, 1809) and Mythimna separata Walker, 1865 (Lepidoptera: Noctuidae) 

Response: done

Thank you for your comments.

We have revised as suggested. (Lines 87-88, page 2 in the marked-up version)

line 101: Acer negundo Linnaeus, 1753

Response: done

Thank you for your comments.

We have deleted this sentence as suggested by reviewer 1. (Lines 102-103, page 2 in the marked-up version)

line 115: Salix matsudana umbraculifera Rehder, 1949 and S. babylonica Linnaeus, 1753

Response: done

Thank you for your comments.

We have revised as suggested. (Line 118, page 4 in the marked-up version)

line 344: Luminance 

Response: done

Thank you for your comments.

We have revised as suggested. (Line 178, page 5 in the marked-up version)

line 344: P value 

Response: done

Thank you for your comments.

We have deleted “value” as suggested. (Line 344, page 10 in the marked-up version)

line 346: Where is "bold type"?. 

Response: done

Thank you for your comments.

We have added “bold type” in the table 2. (Lines 344-345, pages 10-11 in the marked-up version)

line 387: Arhopalus ferus (Mulsant, 1839) (Coleoptera: Cerambycidae)

Response: done

Thank you for your comments.

We have revised as suggested. (Lines 404-405, page 13 in the marked-up version)

line 448: Dastarcus helophoroides (Fairmaire, 1881) (Coleoptera: Bothrideridae) 

Response: done

Thank you for your comments.

We have revised as suggested. (Lines 474-475, page 14 in the marked-up version)

line 450: Monochamus alternatus Hope, 1842 (Coleoptera: Cerambycidae)

Response: done

Thank you for your comments.

We have revised as suggested. (Line 477, page 14 in the marked-up version)

line 476: Drosophila suzukii (Matsumura, 1931) (Diptera: Drosophilidae)

Response: done

Thank you for your comments.

We have revised as suggested. (Line 504, page 15 in the marked-up version)

line 479: Halyomorpha halys (StaÌŠl, 1855) (Heteroptera, Pentatomidae)

Response: done

Thank you for your comments.

We have revised as suggested. (Line 507, page 15 in the marked-up version)

lnie 507-508: 2. Beketov; M.A.; Kefford; B.J.; Schäfer; R.B.; Liess; M. Pesticides reduce regional biodiversity of stream invertebrates. Proc. Natl. Acad. Sci. USA. 2013. 110 (27) 11039-11043 https://doi.org/10.1073/pnas.1305618110

Response: done

Thank you for your comments.

We have revised as suggested. (Line 537, page 16 in the marked-up version)

The other comment

In addition, we found the contributions of some authors are not sufficient. 
Please add other contributions and make sure each author has at least TWO
contributions.

Response: done

Thank you for your comments.

We have revised as suggested. (Lines 519-522, page 15 in the marked-up version)

***************************************

We are grateful for these invaluable comments from the subject editor and reviewers. Thank you very much!

Reviewer 2 Report

Dear authors,

In your paper is presented result of the light trap attraction study to collect Cerambycidae species adults, and it is very interesting to those who has experience in collecting invasive insects in forests. You are absolutely right when writing about difficulties to collect invasive species using traditional traps. Pheromone traps are working on narrow spectrum of species, light traps just occasionally attract flight beetles, canopy traps very hard to mount. That is why your test of different light wavelength lamp attraction to collect longhorn beetle species is actual at present time.

Several questions appeared after acquaintance with your work. 

— beetles living in nature adapted to natural source light, such as day light (380 – 780 nm) or fire (589 nm). Your lighting were 365 nm, 420 nm, 435 nm and 515 nm. Why did not used standard lamps too? Why spectrum of fire has not been tried? 

— what do you think why violet spectrum (420 nm and 435 nm) is more efficient to attract the beetle adults?

— To compare catching efficiency of different trap types the control trapping should be held. For example, different type traps, such as: pheromone, canopy, flight-break etc. should have been applied and result compared with your light experiment 

All these questions are proposed to enhance your future study, the manuscript could be published at present version.

With the kindest wishes.

Author Response

(The authors gave the same response as above.)

Reviewer 3 Report

Recommended amendments and adjustments:

At the first mention of the Latin name, it is necessary to give the full name of the species with the author and year of description. I recommend doing this in the title of the article. 
Research on photoinduction-based technology for trapping Asian longhorned beetle (Anoplophora glabripennis (Motschulsky, 1853) (Coleoptera: Cerambycidae)

line 51: Anoplophora glabripennis delete (Motschulsky)

line 56: delete adult, because the larva is the harmful stage.

line 86: Helicoverpa armigera (Hübner, 1809) and Mythimna separata Walker, 1865 (Lepidoptera: Noctuidae) 

line 101: Acer negundo Linnaeus, 1753

line 115: Salix matsudana umbraculifera Rehder, 1949 and S. babylonica Linnaeus, 1753

line 344: Luminance 

line 344: P value 

line 346: Where is "bold type"?. 

line 387: Arhopalus ferus (Mulsant, 1839) (Coleoptera: Cerambycidae)

line 448: Dastarcus helophoroides (Fairmaire, 1881) (Coleoptera: Bothrideridae) 

line 450: Monochamus alternatus Hope, 1842 (Coleoptera: Cerambycidae)

line 476: Drosophila suzukii (Matsumura, 1931) (Diptera: Drosophilidae)

line 479: Halyomorpha halys (StaÌŠl, 1855) (Heteroptera, Pentatomidae)

lnie 507-508: 2. Beketov; M.A.; Kefford; B.J.; Schäfer; R.B.; Liess; M. Pesticides reduce regional biodiversity of stream invertebrates. Proc. Natl. Acad. Sci. USA. 2013. 110 (27) 11039-11043 https://doi.org/10.1073/pnas.1305618110

Author Response

(The authors gave the same response as above.)

Round 2

Reviewer 1 Report

Accept in present form